# Effects of Deoxynivalenol Detoxifier on Growth Performance, Blood Biochemical Indices, and Microbiota Composition of Piglets

**DOI:** 10.3390/ijms26052045

**Published:** 2025-02-26

**Authors:** Luyao Zhang, Yongwei Wang, Weiwei Wang, Li Wang, Jingjing Shi, Junlin Cheng, Jing Zhang, Aike Li, Beibei He, Zhiyong Fan

**Affiliations:** 1Academy of National Food and Strategic Reserves Administration, Beijing 100037, China; 17835714884@163.com (L.Z.); wyw@ags.ac.cn (Y.W.); www@ags.ac.cn (W.W.); wl@ags.ac.cn (L.W.); sjj@ags.ac.cn (J.S.); cjl@ags.ac.cn (J.C.); zj@ags.ac.cn (J.Z.); lak@ags.ac.cn (A.L.); 2College of Animal Science and Technology, Hunan Agricultural University, Changsha 410215, China

**Keywords:** deoxynivalenol, composite detoxifier, growth performance, blood biochemical indices, microbiota composition, piglets

## Abstract

Deoxynivalenol (DON), also known as vomitoxin, has a high detection and exceeding rate in feed and is prone to causing symptoms such as loss of appetite, weight loss, vomiting, and diarrhoea in animals, which brings great harm to the aquaculture industry. The common mycotoxin adsorbents have low adsorption rates for DON, and the use of biological methods to remove DON in feeds has gradually become a research trend. One hundred and twenty crossbred barrows were randomly divided into four groups, which included the normal diet group (CON), normal diet + detoxifier group (Det), DON-polluted diet group (DON), and DON-polluted diet + DON detoxifier group (DON + Det); the experiment lasted for 28 d. The results showed that, compared with piglets fed a normal diet, those piglets fed DON-polluted diets significantly decreased their average daily gain (ADG) and average daily feed intake (ADFI) during the 1–14 d and 1–28 d periods; the content of immunoglobulin G (IgG), catalase (CAT), superoxide dismutase (SOD), glutathione peroxidase (GSH-PX), interleukin-4 (IL-4), and interleukin-10 (IL-10) in serum was decreased; and the content of aminotransferase (AST), alanine aminotransferase (ALT), malondialdehyde (MDA), diamine oxidase (DAO), and endotoxin (LPS) was increased in pigs fed DON-polluted diets; meanwhile, feeding piglets DON-polluted diets significantly reduced the levels of acetic acid, propionic acid, and total short-chain fatty acids (SCFAs) as well as gut microbiota health index (GMHI) in piglet faeces, but increased the relative abundance of *Treponema*, *Prevotellaceae_*UGG-001, *Lachnospiraceae*_XPB1014_group, *Frisingicoccus* and *Sphaerochaeta*. In contrast, the addition of a composite detoxifier effectively ameliorated the reduction in ADG and ADFI in piglets caused by DON-polluted diets. It suppressed the reduction in CAT, SOD, GSH-PX, IL-4, and IL-10 and the elevation of TNF-α, IL-2, IL-6, IL-12, MDA, LPS, and DAO in serum; the composite detoxifier also restrained the decrease in SCFA in piglet faeces and increased the relative abundance of *Ruminococcus*, *Lachnospiraceae*_NK4A136_group, *Lachnospiraceae*_AC2044_group, *UCG*-009, and *Eubacterium*_siraeum_group bacteria. The composite detoxifier effectively mitigated the adverse effects of a DON-polluted diet on piglet growth performance, blood biochemical indices, and gut microbiota composition.

## 1. Introduction

Mycotoxins are toxic secondary metabolites produced by moulds which are widely present in grain and feed, posing a great threat to the health of humans and animals, and bringing substantial economic losses to the animal husbandry and feed industries [1]. Deoxynivalenol (DON) is a relatively common toxin with a high detection and exceeding rate in animal feed. As reported in previous studies [2,3], DON can cause oxidative stress, inflammation, apoptosis, and damage to the integrity of the intestinal barrier. The detoxification methods for DON mainly include physical, chemical, and biological methods. Among them, biological detoxification technology has emerged as an important approach because of its features such as environmental friendliness, mild reaction conditions, and high specificity and efficiency [4,5]. According to its detoxification mechanism, biological detoxification is mainly classified into biosorption and biodegradation [6,7]. Biosorption detoxification, as a major form of biological detoxification, utilises the adsorption properties of biological materials to remove DON, with yeast being the dominant agent [8]. Some lactic acid bacteria also have a certain adsorption effect on DON. For instance, the cell wall of *Lactobacillus paracasei* can form a complex combination with DON, reducing the content of DON in a reaction solution by 40.7% within 24 h [9]. Biodegradation, on the other hand, mainly employs microorganisms or the enzymes they produce to structurally modify or destroy the C-12,13 epoxy group and C3-OH group, which are the main toxic groups of DON, and convert DON into non-toxic or low-toxic substances. For example, Gao et al. discovered a strain of Gram-positive *non-bacteriophage Slackia* sp. *D-G6* in the chicken intestine, which could completely convert 25 mg/L DON to DOM-1 within 48 h in culture medium [10].

The main toxic groups of DON are the C-12,13 epoxy group and C3-OH group, which are resistant to high temperature and acid, making it difficult to destroy them by simple processing. Therefore, the removal of vomitoxin from feeds has always been a research difficulty [11,12,13]. However, due to the limitations of microbial detoxification products, such as the existence of a single detoxification site, imperfect safety evaluations, and harsh reaction conditions, few of them can be applied to practical use [14,15]. Currently available detoxification agents mainly include chemical reagents containing chelating components, bentonite clay, algae, enzymes, and fungal extracts. Their detoxification effect is not ideal; moreover, they may also adsorb nutrients such as vitamins, amino acids, and trace elements, and they lack systematic safety evaluation [16,17,18]. Recharla et al. [19] showed that in order to effectively utilise mycotoxin-degrading microorganisms as additives in animal feed, certain prerequisites have to be fulfilled. These include the rapid degradation of mycotoxins to non-toxic metabolites under varying oxygen conditions and in complex environments, safety of use, and stability in the gastrointestinal tract at different pH values. Therefore, combining biological and physical methods to improve the efficiency of vomitoxin removal and developing technologies with advantages such as a high efficiency, broad-spectrum application, and environmental friendliness for vomitoxin removal and application are the current development trends.

Holanda et al. added a mycotoxin-detoxifying agent composed of bentonite clay, enzymes, algae, and yeast to the diet of weaned piglets (which containing 2 mg/kg DON). They found that it could reduce the damage to the intestines and liver of the piglets induced by DON [6]. Xu et al. showed that the addition of a *Saccharomyces cerevisiae*-based detoxifier to DON-polluted diets decreased serum aminotransferase (AST), aminotransferase (ALT), and Lactate Dehydrogenase (LDH) concentrations and significantly improved the intestinal damage caused by DON in piglets [20]. Li et al. showed that the addition of *Clostridium butyricum WJ06* to DON-polluted diets changed the structure of the intestinal flora of weaned piglets [21]. In addition, the addition of a composite detoxifier consisting of *Lactobacillus casei*, *Enterobacter faecalis*, *Clostridium perfringens*, *Brettanomyces cerevisiae*, and *Flavobacterium* can improve the abundance and diversity of the intestinal flora of weaned piglets [22]. Fuller et al. showed that adding a composite detoxifier to DON-polluted diets could inhibit the proliferation of *Escherichia coli* and promote the growth of *Lactobacillus lactis* in the intestines of piglets [23].

Pigs are significantly more sensitive to DON than other animals, and it is difficult to remove DON from feed using a single method. Given the importance of ensuring feed safety and animal health in the swine industry, this study aimed to investigate the effects of a compound detoxifier, composed of probiotics, degrading enzymes, and adsorbents, on the growth performance, serum biochemical indices, and intestinal health of piglets fed a DON-polluted diet.

## 2. Results

### 2.1. Growth Performance

As shown in Table 1, there was no significant difference (*p* > 0.05) in the initial body weights of piglets in each group. Compared with the control group, piglets fed DON-polluted diets had lower (*p* < 0.05) body weights at 14 d and 28 d and lower (*p* < 0.05) average daily gain (ADG) at 1–14 d and 1–28 d, as well as lower (*p* < 0.05) average daily feed intake (ADFI) at 1–14 d, 25–28 d, and 1–28 d and significantly higher (*p* < 0.05) F/G at 1–14 d. Compared with the Det group, the DON + Det group significantly decreased in ADG (*p* < 0.05) at 1–28 d and ADFI (*p* < 0.05) at 15–28 d and 1–28 d in piglets. Compared with the DON group, the DON + Det group had a significantly increased ADG (*p* < 0.05) and ADFI (*p* < 0.05) and decreased (*p* < 0.05) F/G (*p* < 0.05) at 1–14 d and 1–28 d in piglets.

### 2.2. Serum Biochemical Indices

Table 2 depicts the effect of DON and the detoxifier on the serum biochemical indices of the piglets. As shown in Table 2, the serum levels of aminotransferase (AST) and aminotransferase (ALT) in piglets fed DON-polluted diets were higher (*p* < 0.05) compared to the control group. The serum levels of AST in piglets in the DON + Det group were higher (*p* < 0.05) compared to the Det group. The serum levels of AST and ALT in the DON + Det group were lower (*p* < 0.05) compared to the DON group. The serum levels of albumin (ALB), globulin (GLB), and total protein (TP) in the DON and DON + Det groups was not significantly different from those in the other groups (*p* > 0.05).

### 2.3. Serum Immunoglobulin

Table 3 depicts the effect of DON and the detoxifier on the serum immunoglobulins of the piglets. As shown in Table 3, compared with the CON group, the immunoglobulin G (IgG) content in the serum of piglets fed with DON-polluted diets was significantly reduced (*p* < 0.05). Compared with the Det group, the DON + Det treatment significantly reduced the IgG content in the serum of piglets (*p* < 0.05). Compared with the DON group, the DON + Det treatment significantly increased the IgG content in the serum of piglets (*p* < 0.05). In addition, the DON and DON + Det treatments had no significant effects on the serum levels of immunoglobulin A (IgA) and immunoglobulin M (IgM) in piglets.

### 2.4. Serum Antioxidant Indices

Table 4 depicts the effect of DON and the detoxifier on the serum antioxidant indices of the piglets. As shown in Table 4, compared with the CON group, the contents of catalase (CAT), superoxide dismutase (SOD), and glutathione peroxidase (GSH-PX) in the serum of piglets fed with DON-polluted diets were significantly decreased (*p* < 0.05). Compared with the DON group, the contents of CAT, SOD, and GSH-PX in the serum of piglets in the DON + Det group were significantly increased (*p* < 0.05). Table 4 indicates that, compared with the CON group, the content of malondialdehyde (MDA) in the serum of piglets in the DON group was significantly increased (*p* < 0.05). Compared with the DON group, the content of MDA in the serum of the piglets in the Don + Det group was significantly decreased (*p* < 0.05).

### 2.5. Serum Inflammatory Factors

Table 5 depicts the effect of DON and the detoxifier on the serum inflammatory factors of the piglets. As shown in Table 5, compared with the CON group, the serum levels of anti-inflammatory factors interleukin-4 (IL-4) and interleukin-10 (IL-10) in piglets fed with the DON-polluted diets were significantly decreased (*p* < 0.05), while the levels of pro-inflammatory factors interleukin-12 (IL-12), interleukin-2 (IL-2), and interleukin-6 (IL-6) were significantly increased (*p* < 0.05); compared with the DON group, the serum levels of anti-inflammatory cytokines IL-4 and IL-10 in piglets from the DON + Det group were significantly increased (*p* < 0.05), and the levels of pro-inflammatory factors tumour factor-α (TNF-α), IL-12, IL-2, and IL-6 were significantly decreased (*p* < 0.05).

### 2.6. Endotoxin and Diamine Oxidase

As shown in Table 6, the serum levels of lipopolysaccharide (LPS) and diamine oxidase (DAO) in piglets fed DON-polluted diets were significantly higher (*p* < 0.05) compared to the CON group, while the serum levels of LPS and DAO in piglets in the DON + Det group were significantly lower (*p* < 0.05) compared to the DON group. Additionally, there were no significant differences in the contents of LPS and DAO in the serum of piglets in the DON + Det group compared with the Det group (*p* > 0.05).

### 2.7. Short-Chain Fatty Acids

Figure 1 is a diagram depicting the effect of DON and the detoxifier on short-chain fatty acids in piglet faeces. As shown in Figure 1, the contents of acetic acid, propionic acid, and total SCFAs in the faeces of piglets fed DON-polluted diets were significantly lower (*p* < 0.05) compared to the CON group. Compared with the Det group, the contents of acetic acid, propionic acid, and total SCFAs in the faeces of piglets in the DON + Det group were significantly decreased (*p* < 0.05). And the contents of acetic acid, propionic acid, and total SCFAs in the faeces of piglets in the DON + Det group were significantly higher (*p* < 0.05) compared to the DON group. In addition, there was no significant difference (*p* > 0.05) in the faecal content of lactic acid and butyric acid between the DON and DON + Det groups.

### 2.8. Faecal Microbiota Composition

#### 2.8.1. Diversity Analysis

As shown in Figure 2A, the constructed dilution curves all tended to be flat, indicating that the sequencing data had reached saturation and that the sequencing depth was reasonable. The richness and diversity of bacterial communities were measured by diversity indices. As shown in Figure 2B, the Chao and Shannon indices of piglet faeces flora were not significantly different between each group (*p* > 0.05). The PCoA analysis shows that the composition of faeces microbiota for the piglets in each group was clearly differentiated and formed clear clusters (Figure 2C). The number of OTUs in the faeces microbiota of the piglets in the CON, Det, DON, and DON + Det groups were 1440, 1331, 1608, and 1729, respectively, with a total of 924 OTUs in the four groups, and the numbers of unique OTUs were 128, 104, 260, and 252, respectively (Figure 2D). The gut microbiome health index (GMHI) is a health status assessment tool based on the gut microbiome taxonomic profile used to predict whether an individual has a clinically diagnosed disease. As shown in Figure 2E, the GMHI of the faecal microbiota was significantly lower (*p* < 0.05) in piglets fed DON-polluted diets compared with the normal diet group.

#### 2.8.2. Taxonomic Composition

Figure 3 is a diagram depicting the effect of DON and the detoxifier on the microbiota composition in the piglets’ faeces. At the phylum level, the microbiota of the piglets’ faeces were composed of Firmicutes, Bacteroidetes, Spirochaetaes, Actinobacteria, Proteobacteria, Desulfobacterota, Cyanobacteria, WPS-2, Verrucomicrobiota, and Patescibacteria, accounting for more than 99% of the whole flora, of which Firmicutes, Bacteroidetes, and Spirochaetaes are the dominant bacteria in faeces (Figure 3A).

At the genus level, OTUs were dominated by *Clostridium*_sensu_stricto_1, *Treponema*, *Lactobacillus*, *Prevotella*, *norank_f_Muribaculaceae*, *Streptococcus, Selenomonas*, *Terrisporobacter*, *Dialister*, *Christensenellaceae*_R-7_group, *Prevotellaceae*_NK3B31_group, *norank_f_norank_O_Clostridia_UCG-014*, *Megasphaera, UCG-002*, and *UCG-005* (Figure 3B).

According to the obtained community abundance data, the Kruskal–Wallis H test was used to test the hypothesis of species differences between different groups of microbial communities, evaluate the significance level of species abundance differences, and obtain the species with significant differences between groups (Figure 3C). At the phylum level, the abundance of Firmicutes was significantly decreased (*p* < 0.05) and the abundance of Spirochaetae were significantly increased in the faeces of piglets in the DON group compared with the Det group (*p* < 0.05). Compared with the control group, the abundance of Bacteroidetes in the faeces of piglets in the Det group was significantly decreased (*p* < 0.05). At the class level, Spirochaetae in faeces of piglets in DON group were significantly increased compared with the Det group (*p* < 0.05). Compared with the control group, the number of Bacteroidetes in the faeces of piglets in the Det group was significantly decreased (*p* < 0.05). At the order level, the number of Bacteroidetes in the faeces of piglets in the Det group was significantly decreased compared with the control group (*p* < 0.05). Compared with the Det group, the Spirochaetae in the faeces of piglets in the DON group was significantly increased (*p* < 0.05). At the family level, the Spirochaetae in the faeces of piglets in the DON group was significantly increased compared with the Det group (*p* < 0.05).

#### 2.8.3. Analysis of Differences at Genus Level of Faeces Flora of Piglets

Based on linear discriminant analysis (LDA), linear discriminant analysis effect size (LEfSe) was used to analyse the significant difference between groups, and LDA thresholds ≥ 2 are listed in Figure 4. There were 8 bacteria with higher abundance in the control group, of which *UCG-005*, *Burkholderia-Caballeronia-Paraburkholderia*, and *Acidaminococcus* were the dominant bacteria (*p* < 0.05); there were 15 bacteria with higher abundance in the detoxifier group, of which *Dialister*, *Subdoligranulum*, *Erysipelotrichaceae_UCG-009*, *Erysipelotrichaceae_UCG-006*, and *Oribacterium* were the dominant bacteria (*p* < 0.05); 11 bacteria were more abundant in the DON group, among which *Treponema*, *Prevotellaceae UCG-001*, and *Lachnospiraceae XPB1014_group* were the dominant bacteria (*p* < 0.05); the DON + detoxifier group contained 6 bacteria with higher abundances, among which *Ruminococcus*, *Lachnospiraceae NK4A136_*group, and *Lachnospiraceae AC2044* group were the dominant bacteria (*p* < 0.05).

#### 2.8.4. Analysis of Correlation Between Faeces Microorganisms and Environmental Factors of Piglets

As shown in Figure 5A, differential abundant microorganisms in genus level were correlated with serum antioxidants (MDA content, SOD, GSH-PX, CAT enzyme activity), inflammatory factors (TNF-α, IL-2, IL-4, IL-6, IL-10, and IL-12), and DAO content. The results showed that the order of magnitude of correlation between environmental factors and differential microorganisms was IL-2 > CAT > DAO > IL-6 > IL-4 > TNF-α > SOD > GSH-PX > IL-12 > MDA > IL-10.

As shown in Figure 5B, the abundance of *Lactobacillus* was positively correlated with the contents of DAO and IL-2 (*p* < 0.05) and negatively correlated with the enzyme activities of SOD and CAT in piglet serum (*p* < 0.05). The abundance of *Treponema* was positively correlated with DAO content in piglet serum (*p* < 0.05). The abundance of *Christensenellaceae*_R-7_group was positively correlated with the content of MDA (*p* < 0.05). The abundance of *Prevotella* was positively correlated with the content of IL-10 and the enzyme activity of CAT and GSH-PX (*p* < 0.05) and negatively correlated with the content of IL-6 and MDA in piglet serum (*p* < 0.05). The abundance of *Dialister* was positively correlated with CAT enzyme activity (*p* < 0.05) and negatively correlated with MDA, DAO, and IL-2 contents in piglet serum (*p* < 0.05). The abundance of *Eubacterium ruminantium* was negatively correlated with the content of TNF-α in serum (*p* < 0.05). The abundance of *UCG-005* was positively correlated with CAT and GSH-PX activity (*p* < 0.05) and negatively correlated with TNF-α content in piglet serum (*p* < 0.05). The abundance of *Ruminococcus* was positively correlated with CAT activity (*p* < 0.05) and negatively correlated with TNF-α content in piglet serum (*p* < 0.05). In addition, there was no significant correlation between the contents of IL-4 and IL-12 in piglet serum and the differential abundant flora (*p* > 0.05).

## 3. Discussion

DON-polluted diets have been reported to cause rejection, vomiting, reduced immune response, and intestinal disturbance in animals [24]. In this experiment, piglets were fed a DON-polluted diet for 28 days. It was found that the ADG and ADFI of those piglets were significantly reduced, which was consistent with the studies of Liu and Liao [25,26]. Due to the poor electrophilicity of the DON molecule, it is difficult to combine with silicaluminate sorbents through charge adsorption. We prepared a composite detoxifier, which was composed of *Pediococcus pentose*, *Saccharomyces cerevisiae*, *Lactobacillus plantarum*, glucose oxidase, adsorbent, and eucommia ulmoides leaf extract. We found that it effectively ameliorated the reduction in ADG and ADFI in piglets caused by DON-polluted diets (Table 1). This indicates that this composite detoxifier can counteract the decrease in the growth performance of piglets caused by DON-polluted diets.

The nutrients consumed by animals are decomposed into small molecule metabolites in the gastrointestinal tract by digestive enzymes. Subsequently, these metabolites enter the blood circulation through the transportation mechanism of intestinal epithelial cells. Therefore, serum biochemical indices can effectively reflect the dynamic changes in nutrient metabolism and physiological functions in animals [3]. It was suggested that the ingestion of feed contaminated with DON by piglets may cause liver damage, which releases AST and ALT from the cytoplasm of hepatocytes into the blood, resulting in a significant increase in serum levels of ALT and AST [27]. In this study, the serum contents of AST and ALT in piglets fed DON-polluted diets were significantly increased. This indicates that this composite detoxifier could effectively alleviate the liver injury in piglets induced by DON (Table 2).

Mycotoxins are one of the important factors that initiate and exacerbate oxidative stress in piglets [27]. In this experiment, the serum levels of CAT, SOD, and GSH-PX in piglets fed DON-polluted diets were significantly reduced and the level of MDA was significantly increased (Table 4), whereas, after the addition of the composite detoxifier, the serum levels of CAT, SOD, and GSH-PX were significantly increased and the level of MDA was significantly decreased, which is consistent with the study of Li et al. [21]. Previous research has demonstrated that DON induces the activation of NF-κB, triggers the inflammatory response, and selectively induces the up-regulation of the expression of a series of cytokines, chemokines, and other immune-related inflammatory factors. In addition to the above inflammatory factors, the results of this study showed that feeding DON-polluted diets also significantly increased the serum levels of pro-inflammatory factors TNF-α, IL-12, IL-2, and IL-6 in piglets (Table 3). In addition, a study showed no significant changes in serum levels of IgA, IgG, and IgM in piglets after 4 weeks of feeding them a diet with a DON content of 1.2 mg/kg [28]. In another study, after feeding piglets a diet with a DON content of 4 mg/kg for 4 weeks, there was no change in the serum levels of IgA, but the levels of IgM and IgG were elevated [29]. However, in the study by Pinton et al., the serum levels of IgA and IgG were significantly increased in pigs fed DON (2.2–2.5 mg/kg)-polluted diets [30,31]. In the present experiment, only a decrease in serum IgG levels was observed in piglets fed the DON-polluted diet, while no significant changes were observed in the levels of IgA and IgM. The discrepancies in these results may be attributed to factors such as the amount of DON in the diet, the sex of the animals, the initial level of immunoglobulins, and different feeding methods. In conclusion, feeding piglets DON-polluted diets compromises both their antioxidant and immune functions. However, the addition of a composite detoxifier to DON-polluted diets can alleviate oxidative stress. It does so by enhancing the body’s antioxidant capacity through increasing the contents of CAT, SOD, and GSH-PX, and by reducing the expression of pro-inflammatory factors such as IL-12, IL-2, and IL-6.

Intestinal permeability denotes the characteristic of the intestinal wall to selectively permit certain substances to pass in and out between the intestinal lumen and tissues [16], and it is also a crucial indicator of intestinal barrier function. In animals, elevated serum levels of LPS and DAO, two biomarkers indicative of intestinal permeability, signify increased intestinal permeability and compromised intestinal barrier function [32,33]. DON-polluted diets cause a significant increase in serum DAO levels in piglets [33]. Feng et al. demonstrated that the exogenous addition of detoxification agents with sodium butyrate as the main ingredient was capable of reducing serum levels of LPS and DAO and effectively alleviating diarrhoea in weaned piglets [23]. Van found that the serum levels of LPS and DAO in piglets fed DON-contaminated diets were significantly increased, while both levels were significantly reduced after the addition of a compound detoxifier [34]. The above study is consistent with the results of the present experiment. The serum levels of LPS and DAO in piglets fed DON-polluted diets increased significantly. In contrast, the addition of the composite detoxifier significantly reduced the serum levels of LPS and DAO in piglets, restoring them to their normal levels (Table 6). This suggests that the composite detoxifier can effectively mitigate the alteration of porcine intestinal permeability induced by DON.

The intestinal flora plays a key role in animal health and disease defence, and dysbiosis of the intestinal flora has been linked to an array of diseases, such as inflammatory bowel disease, obesity, and colorectal cancer [35]. In this experiment, the addition of DON and a composite detoxifier to the diet had no significant effect on the α-diversity index of the faecal flora in piglets. However, when the β-diversity, which focuses on the differences in community composition between samples, was combined with a Venn diagram analysis, it was revealed that the DON-polluted diet and the addition of the composite detoxifier altered the structure of the faecal flora in piglets (Figure 2).

To better compare the effects of DON and the combined detoxifier on the faecal flora structure of piglets, bacteria with an average relative abundance greater than 0.01% in all samples were selected for analysis. In this study, the intestinal microbiota of piglets at the phylum level was mainly composed of Firmicutes and Bacteroidetes, which is in agreement with the findings of most studies [36]. Moreover, several studies have shown that Firmicutes play an important role in energy metabolism and the maintenance of intestinal health in the body [37]. The abundance of Firmicutes was significantly lower in the luminal contents of the ileum, cecum, and colon of piglets fed DON-polluted diets, yet no significant difference was observed in the faeces of these piglets [38]. Bacteroidetes are one of the most important commensal organisms in the gastrointestinal flora of mammals and is capable of inhibiting the proliferation of harmful bacteria in the host’s intestine. Scaldaferri et al. found that inflammation in the human gut resulted in a significant decrease in the abundance of Bacteroidetes [39]. A study showed that the abundance of Bacteroidetes was significantly reduced in mice consuming DON-polluted diets, which is consistent with the results of the present study, suggesting that DON may alter the composition of the faecal flora of piglets mainly by affecting the abundance of the bacteria belonging to the phylum Bacteroidetes [40]. In addition, Zhai et al. showed that the abundance of Spirochaetaes in the faeces of laying hens fed DON-polluted diets was significantly higher, and it showed an increasing trend with an increase in DON concentration [41]. In the present study, DON-polluted diets also increased the abundance of Spirochaetaes at the phylum, order, and family levels. However, after the addition of the composite detoxifier, the abundance of Spirochaetaes decreased and returned to the same level as that of the control group.

At the genus level, *Treponema*, *Prevotellaceae*_UGG-001, *Lachnospiraceae_XPB1014_group*, and *Firsingicoccu*s were the dominant organisms in the faeces of piglets in the DON group. *Treponema* is the causative agent of porcine dysentery, a form of mucous haemorrhagic diarrhoeal disease in weaned piglets. The feeding of DON-polluted diets increased the abundance of *Treponema* bacteria in piglets’ faeces in this experiment, while the abundance of *Treponema* was significantly reduced after the addition of the composite detoxifier, which is consistent with the findings of Zhang and the finding that the detoxifier can alleviate piglet diarrhoea by inhibiting the growth and colonisation of *Treponema* [42]. Moreover, current studies generally agree that *Prevotella* is beneficial to intestinal homeostasis and health, mainly related to intestinal nutrition and metabolism, by degrading dietary fibre in plant cell walls and producing large amounts of SCFAs for host absorption [43]. Some studies have shown that AFB1-polluted diets can lead to a reduction in the abundance of *Prevotellaceae_UGG-001* bacteria in the rumen microflora of sheep, which is consistent with the findings of the present study, a reduction in the abundance of *Prevotellaceae_UGG-001* bacteria in the faeces of piglets consuming DON-polluted diets. However, another study found that the abundance of *Prevotellaceae_UGG-001* bacteria in the cecum of piglets increased after 28 d of feeding diets containing 2.89 mg/kg DON, which was contrary to the results of the present study, probably because the intestinal flora has a high degree of dynamics and the number of some microorganisms may change with the different diets fed and with the lapse of time of treatment [44]. Therefore, the relationship between the content of DON in the diet, feeding time, and the dynamics of some dominant bacteria in the intestinal flora such as *Prevotellaceae_UGG-001* needs to be clarified in the future. In addition, Jin et al. showed that the abundance of *Lachnospiraceae_XPB1014_group* was significantly increased in the cecum of mice fed DON-polluted diets [45]. Wang et al. showed that the abundance of the bacterium *Lachnospiraceae_XPB1014_group* in the faeces of piglets fed DON-polluted diets was significantly increased; after the addition of the composite detoxifier, the abundance of this bacterium decreased, which is consistent with the results of the present study [46]. One study showed that the abundance of *Firsingicoccus* bacteria in the colon of rats with spleen deficiency and diarrhoea was significantly increased, while the addition of Pingweisuan alleviated the diarrhoea of rats by reducing the abundance of *Firsingicoccus* [47]. Feng et al. investigated the changes in the intestinal microbial composition of patients with spinal muscular atrophy and found that the abundance of *Firsingicoccus* in the colon of the patients was significantly higher than that of the in the normal population [48]. Although no study has yet reported a direct relationship between *Firsingicoccus* and DON, this study for the first time revealed that the abundance of *Firsingicoccus* in the faeces of piglets fed DON-polluted diets was significantly increased. Simultaneously, the addition of a compound decontamination agent significantly reduced the abundance of Firsingicoccus in the faeces of piglets. However, whether this bacterium plays a key role in the toxic effects induced by DON needs to be further confirmed.

At the genus level, *Ruminococcus*, *Lachnospiraceae NK4A136_group*, and *Lachnospiraceae AC2044* group were the dominant bacteria in the faeces of pigs in the DON + Det group. Among them, *Ruminococcus*, a genus of bacteria capable of producing abundant propionic acid and butyric acid, is widely involved in the body’s food digestion, intestinal barrier maintenance, and other functions [49]. Bai et al. found that the abundance of *Ruminococcus* in the intestinal tract of piglets fed DON-polluted diets was reduced, and it decreased in the intestinal tracts of piglets after adding a composite detoxifier with *Lactobacillus rhamnosus* as the main ingredient [50]. Wu et al. found that the addition of *Lactobacillus plantarum* as the main component of a composite detoxifier to DON-polluted diets significantly increased the abundance of *Ruminococcus* in the intestinal tract of broilers [51]. In the present study, after the addition of a composite detoxifier to DON-polluted feed, the abundance of *Ruminococcus* in piglet faeces was significantly increased, and the content of propionic acid in piglet faeces was significantly increased, which is consistent with the findings of the above studies (Figure 4). Studies have shown that *Lachnospiraceae* bacteria can produce butyrate, which can enhance the integrity of intestinal epithelial cells, inhibit the body’s inflammatory response, and play an important role in maintaining the health of the body’s gastrointestinal tract [52]. Wu et al. investigated the changes in the intestinal microbial composition of mice with dextrose sodium sulphate-induced ulcerative colitis and found that in the intestinal tracts of mice with colitis, the abundance of *Lachnospiraceae NK4A136_group* was significantly lower than that of normal mice, while the abundance of *Lachnospiraceae NK4A136_group* in the intestinal tract of mice was increased by adding baicalein polysaccharide to the mouse diet [53]. Robie et al. added a variety of probiotic bacteria to the feed of growing pigs and found that the growing pigs had *Lachnospiraceae NK4A136_group* in their faeces [54]. Feng et al. showed that the abundance of *Lachnospiraceae AC2044* group bacteria in the intestinal microorganisms of spinal muscular atrophy patients was significantly lower than that of the normal population [48]. Zhang et al. added *Lactobacillus mucilaginousus citriodora* to the diets of growing pigs and found that the abundance of *Lachnospiraceae AC2044* group bacteria in the intestinal microbiota of growing pigs was significantly lower than that of the normal population. *Lachnospiraceae AC2044* group bacteria increased in abundance [55]. In the present study, after the addition of a composite detoxifier to DON-polluted diets, the abundance of *Lachnospiraceae NK4A136_*group bacteria and *Lachnospiraceae AC2044* group bacteria in piglet faeces increased significantly, which is in line with the results of the above studies.

By studying the relationship between microflora and environmental factors in piglet faeces, the intrinsic relationship between faecal microflora and DON metabolism, as well as animal physiological performance, was further explored. Wang et al. [56] added sodium dextran sulphate to mouse diets and found that the abundance of Prevotella in the faeces of mice decreased. In contrast, in the present study, Prevotella was significantly negatively correlated with the levels of IL-6 and MDA. This indicates that DON exerts a toxic effect on the intestine, elevating the levels of IL-6 and MDA and triggering an inflammatory response in the intestine of piglets. In this study, the abundance of Lactobacillus in piglet faeces was significantly positively correlated with the serum levels of DAO and IL-2 and significantly negatively correlated with the enzyme activities of SOD and CAT in serum; the abundance of Treponema was significantly positively correlated with the serum level of DAO. Furthermore, the abundance of Prevotella was significantly positively correlated with the serum IL-10 content, and the enzyme activities of CAT and GSH-PX were significantly positively correlated with serum IL-10 content and significantly negatively correlated with IL-6 and MDA content (Figure 5).

## 4. Materials and Methods

### 4.1. Animal Ethics Statement

All experiments were conducted in accordance with the Chinese Guidelines for Animal Welfare and Experimental Protocol, and prior approval was obtained from the Animal Care and Use Committee of the Academy of National Food and Strategic Reserves Administration (ethical approval code: 20240814002).

### 4.2. Animal and Experimental Design

The experiment was a two-factor completely randomised group design, with 120 healthy Landrace × Yorkshire (L × Y) barrows with an initial body weight of 17.65 ± 0.20 kg randomly divided into 4 treatment groups, with 6 replicate pens per treatment and 6 pigs per pen. Four dietary treatments were designed in accordance with the additions of mouldy maize and the compound detoxifier, namely, the normal diet group (CON), normal diet + detoxifier group (Det), DON-polluted diet group (DON), and DON-polluted diet + DON detoxifier group (DON + Det). Pigs were housed in pens with drinkers, feeders, and slatted floors, and were provided water and feed freely. The environment temperature was controlled at 22 ± 2 °C. The experiment lasted for 28 days, and deworming and immunisation were carried out according to the routine management procedures.

### 4.3. Composition and Chemical Analysis of Diets

DON-polluted maize (2500 μg/kg) was purchased from Muyuan Food Co., Ltd. (NanYang, China). The composite detoxifier was composed of *Pediococcus pentose* WT-1, *Saccharomyces* cerevisiae JM-2, *Lactobacillus plantarum* ZW-1, glucose oxidase, adsorbent, and Eucommia ulmoides leaf extract in equal proportion. The three strains were preserved and produced by the Academy of National Food and Strategic Reserves Administration (*Pediococcus pentosus* WT-1, 3.5 × 10^12^ CFU/g; *Lactobacillus plantarum* ZW-1, 3.0 × 10^12^ CFU/g; *Saccharomyces cerevisiae* JM-2, 2.5 × 10^10^ CFU/g). Glucose oxidase (10,000 U/g) and eucommia ulmoides leaf extract were purchased from Shanghai Yuanye Biotechnology Co., Ltd (Shanghai, China). Sorbent was purchased from Hunan Taikangmei Biotechnology Co., Ltd (Changsha, China).

All diets were formulated to meet the nutritional requirements for weaned piglets recommended by the NRC (2012) (Table 7). The ME and SID lysine, methionine, threonine, and tryptophan in all the diets were kept the same.

### 4.4. Sample Collection and Processing

At the end of the trial, one piglet of average weight was selected from each cage. After overnight fasting, blood samples were collected by anterior vena cava puncture, centrifuged at 1500× *g* for 10 min at room temperature, and serum was immediately stored at −20 °C for biochemical parameters analysis. Fresh faeces were collected as soon as they were discharged, placed in 50 mL centrifuge tubes, quick-frozen in liquid nitrogen and stored at −80 °C, and used for the high-throughput sequencing of microbial 16S rRNAs and determination of the content of short-chain fatty acids (SCFAs).

### 4.5. Growth Performance

At days 1, 14, and 28 of piglet feeding, the body weight (BW) of the piglets was measured after a 12 h fasting period with continuous access to water. The feed intake of the piglets was recorded on a per-cage basis, and the average daily gain (ADG), average daily feed intake (ADFI), and feed-to-gain ratio (F/G) were calculated for the periods of 1–14 days, 15–28 days, and the entire experimental period.

### 4.6. Serum Parameters

Serum samples were thawed at 4 °C and mixed well prior to analysis. Serum biochemical parameters including total protein (TP), albumin (ALB), globulin (GLB), aspartate transferase (AST), and alanine transaminase (ALT) of the piglets were determined using an Automatice Biochemical Autoanalyzer (Beckman CX4, Beckman Coulter Inc., Brea, CA, USA) using commercial kits (Shanghai Kehua Bio-Engineering Co., Ltd., Shanghai, China). The concentrations of malondialdehyde (MDA), catalase (CAT), superoxide dismutase (SOD), and glutathione peroxidase (GSH-PX) were determined using the thiobarbituric acid method. The reagent kits were provided by Nanjing Jiancheng Bioengineering Co., Ltd. (Nanjing, China), and used following the instructions provided in the kits. The contents of diamine oxidase (DAO), lipopolysaccharide (LPS), immunoglobulin A (IgA), immunoglobulin M (IgM), immunoglobulin G (IgG), tumour factor-α (TNF-α), interleukin-4 (IL-4), interleukin-10 (IL-10), interleukin-12 (IL-12), interleukin-2 (IL-2), and interleukin-6 (IL-6) were measured by enzyme-linked immunosorbent assay kits following the manufacturer’s instructions (Jiangsu Meimian Industrial Co., Ltd., Yancheng, China).

### 4.7. Determination of SCFA Content in Faeces

A total of 1 g of faeces was mixed with 2 mL of ultrapure water and homogenised, then centrifuged at 3000 r/min for 10 min. We took 1 mL of the supernatant using a syringe, filtered it through an organic membrane (0.22 μm), and collected the filtrate in a sealed brown glass sample bottle specifically designed for HPLC analysis. This sample was used to measure the concentration of short-chain fatty acids (SCFAs).

A High-Performance Liquid Chromatography (HPLC) method with a UV detector, C18 column—150 mm length, 4.6 mm inner diameter—filled with 5 μm diameter particles or equivalent, the column at room temperature, a mobile phase comprising phosphate buffer (pH 5.5) and 99.9% chromatography grade methanol (phosphate buffer: methanol = 95:5), a flow rate of 1 mL/min, a detection wavelength of 218 nm, and an injection volume of 20 μL was used. After filtering the sample through a 0.45 μm membrane, we injected 1 mL of the filtered sample for analysis.

### 4.8. High-Throughput Sequencing Analysis of 16S rRNA in Faeces Microbiome

Total DNA extraction of faeces microbial genome: 0.5 g of faeces samples were weighed, and the total DNA of the microbiome was extracted by a QIAamp DNA kit; the extraction process was performed referring to the instruction manual of the kit, the step of magnetic bead striking was added, and the extracted DNA was run through agarose gel electrophoresis for quality detection.

PCR amplification of the V3-V4 region of the 16S rRNA sequence and library construction: This step was performed by Shanghai Meiji Biomedical Technology Co. (Shanghai, China)The V3-V4 region of the 16S rRNA gene was amplified by PCR using specific primers with barcodes. To ensure the reliability and accuracy of the amplified sequences, the number of amplification cycles for each sample was ensured to be the same. The PCR-amplified products were then purified, quantified, and homogenised, and sequencing libraries were created. After the libraries passed the quality control check, microbial diversity sequencing analysis was carried out using an Illumina HiSeq 2500.

After splicing and filtering the readings of each sample, OTUs (operational taxonomic units) were clustered using UPARSE (version 7.11, http://drive5.com/uparse/, accessed on 19 February 2024), and chimeric sequences were identified and removed using UCHUIME software, (http://drive5.com/usearch/manual/uchime_algo.html, accessed on 23 February 2024). The phylotype classification analysis of the Silva (SSU123) 16S rRNA database was performed using the RDPv2.2 classification program with a confidence threshold of 70%. Further α-diversity analysis and β-diversity analysis were performed; the 16S rRNA gene-sequencing results were applied to the I-sanger platform to predict species annotation, composition, variation, and the functional prediction of bacterial communities in piglet faeces. Wayne plots were used to evaluate the distribution of the OTUs among different groups. Unweighted UniFrac distances were measured to determine differences in microbial communities. Finally, selected environmental factors were correlated with microorganisms for heat map analysis.

### 4.9. Data Analysis

Data were analysed using the General Linear Model (GLM) procedure in SPSS 26.0 software to perform a two-way analysis of variance considering interaction effects. If significant main effects or interaction effects were found, post hoc tests were conducted using Duncan’s multiple comparisons. The results were presented as the mean ± the SEM, where *p* < 0.05 indicated statistical significance and 0.05 < *p* < 0.1 indicated a trend towards significance. Graphs were generated using GraphPad Prism 9.0.

16S rRNA analysis in microbiomics: QIIME 2.0 software was employed to analyse species α-diversity indices, including the Shannon and Chao-1 indices, to assess species richness and diversity. Based on the phylogenetic relationships among operational taxonomic units (OTUs), principal coordinates analysis (PCoA) using UniFrac distance matrices was conducted to calculate β-diversity between the samples, evaluating similarities among different communities. Through linear discriminant analysis (LDA), the linear discriminant analysis effect size (LEfSe) was utilised for inter-group significance differential analysis, highlighting taxonomic groups with LDA scores ≥ 2, where a higher LDA value indicated greater differences between groups. Spearman correlation analysis was applied to examine the interrelationships between the microbial samples and various environmental factors.

## 5. Conclusions

The compound detoxifier used in this study can regulate flora homeostasis by increasing the abundance of beneficial bacteria and decreasing the abundance of harmful bacteria, so as to alleviate the adverse effects of a DON-contaminated diet on the growth performance, antioxidant function, and inflammatory response of piglets. Looking ahead, further research could explore the optimal formulation of this composite detoxifier and its application effects across different growth stages and feeding environments, aiming to maximise its detoxifying efficacy while reducing production costs. Additionally, conducting in-depth studies on the mechanism of action of this detoxifier, particularly its regulatory pathways at the molecular level, will provide theoretical foundations and technical support for the development of novel and highly efficient feed detoxifiers.

## Figures and Tables

**Figure 1 ijms-26-02045-f001:**
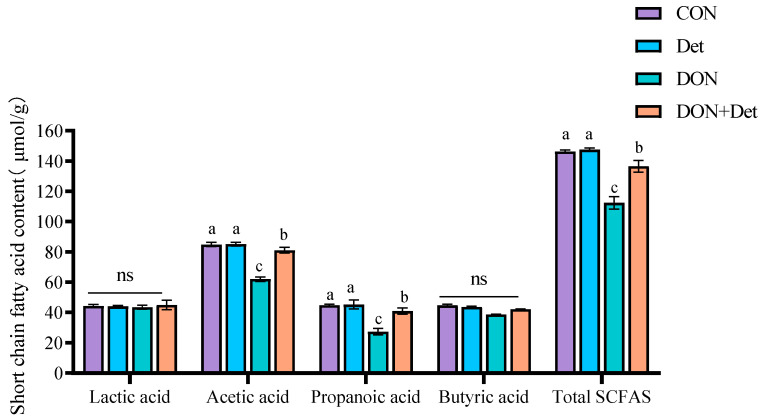
Effects of DON and detoxifier on short-chain fatty acids in piglet faeces. Note: Results are expressed as mean ± SEM, n = 4. Each indicator has different shoulder letters, indicating significant differences (*p* < 0.05).

**Figure 2 ijms-26-02045-f002:**
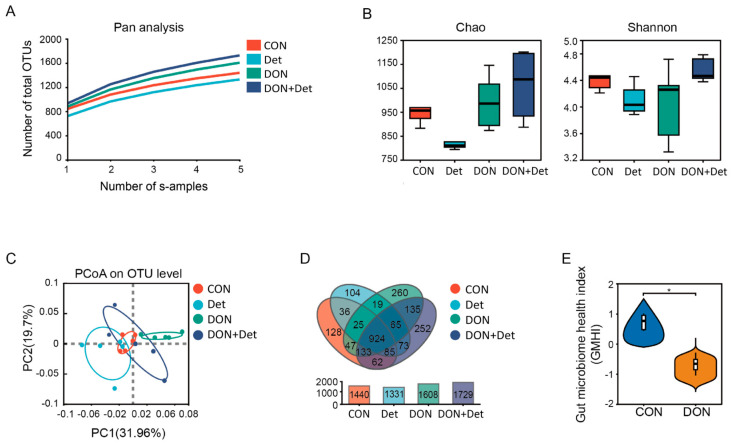
Effects of DON and detoxifier on microbiota diversity in piglets. Note: (**A**) Pan species analysis; (**B**) α diversity indices; (**C**) β diversity analysis; (**D**) Venn diagram; (**E**) health index analysis. * means differ (*p* < 0.05).

**Figure 3 ijms-26-02045-f003:**
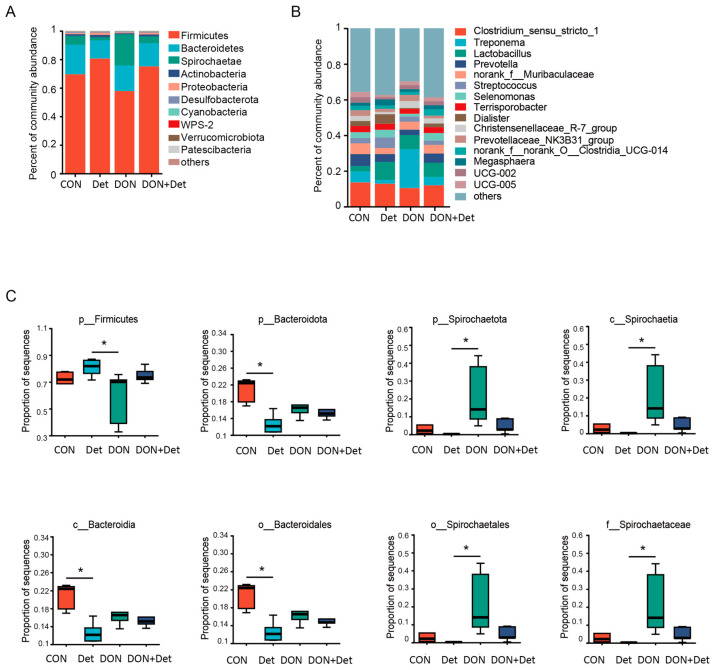
Effects of DON and detoxifier on microbiota composition in piglet faeces. Note: (**A**) Composition of microbiota at phylum level; (**B**) composition of microbiota at genus level; (**C**) differential abundance of bacteria from phylum to family. * means differ (*p* < 0.05).

**Figure 4 ijms-26-02045-f004:**
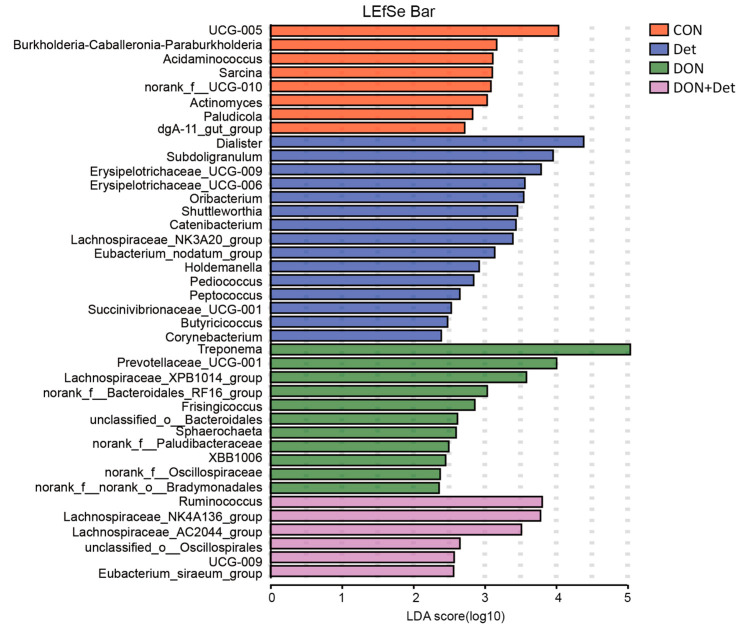
LEfSe analysis of microbiota in piglet faeces. Note: Orange, CON; blue, Det; green, DON; purple, DON + Det.

**Figure 5 ijms-26-02045-f005:**
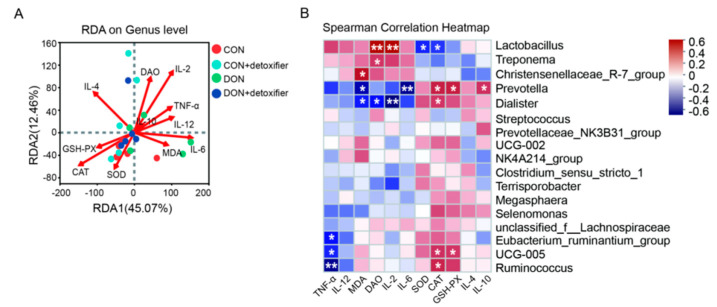
Correlation analysis between differential abundance of bacteria and environmental factors. Note: (**A**) RDA plot of correlation analysis between differential abundance of bacteria and environmental factors; (**B**) heat map of correlation analysis between differential abundance of bacteria and environmental factors. “*” indicates *p* < 0.05 and “**” indicates *p* < 0.01.

**Table 1 ijms-26-02045-t001:** Effects of DON and detoxifier on growth performance of piglets.

Items	Treatments				SEM	*p* Value		
	CON	Det	DON	DON + Det		Det	DON	Det × DON
Initial weight (kg)	17.67 ± 0.06	17.62 ± 0.13	17.61 ± 0.09	17.65 ± 0.11	0.02	0.92	0.76	0.28
14 d weight (kg)	24.75 ± 0.72 ^a^	24.98 ± 0.36 ^a^	23.54 ± 0.59 ^b^	24.24 ± 1.02 ^ab^	0.15	0.12	<0.01	0.42
28 d weight (kg)	33.98 ± 0.70 ^ab^	34.85 ± 1.04 ^a^	32.29 ± 0.92 ^b^	33.25 ± 2.42 ^ab^	0.29	0.14	0.01	0.94
ADG (g/d)								
1–14 d	544.83 ± 52.08 ^a^	565.00 ± 22.58 ^a^	418.33 ± 72.5 ^b^	505.00 ± 82.64 ^ab^	12.63	0.05	<0.01	0.20
15–28 d	658.33 ± 39.2	706.67 ± 70.05	621.67 ± 67.35	641.67 ± 122.05	16.42	0.31	0.14	0.67
1–28 d	603.33 ± 23.38 ^ab^	661.67 ± 37.1 ^a^	526.67 ± 47.19 ^b^	576.67 ± 88.47 ^b^	11.17	0.03	<0.01	0.85
ADFI (g/d)								
1–14 d	875.97 ± 30.89 ^a^	871.97 ± 46.18 ^a^	789.33 ± 49.48 ^b^	879.92 ± 13.53 ^a^	7.72	0.05	0.02	<0.01
15–28 d	1162.79 ± 111.1 ^a^	1237.93 ± 107.2 ^a^	900.38 ± 121.61 ^b^	1005.71 ± 49.69 ^b^	20.69	0.04	<0.01	0.72
1–28 d	1024.69 ± 50.6 ^a^	1061.73 ± 60.66 ^a^	846.91 ± 57.89 ^b^	898.77 ± 33.67 ^b^	10.57	0.05	<0.01	0.73
F/G (g/g)								
1–14 d	1.62 ± 0.13 ^b^	1.54 ± 0.07 ^b^	1.94 ± 0.40 ^a^	1.78 ± 0.28 ^ab^	0.05	0.27	0.01	0.68
15–28 d	1.77 ± 0.16	1.74 ± 0.09	1.63 ± 0.22	1.44 ± 0.26	0.04	0.19	0.01	0.31
1–28 d	1.70 ± 0.12	1.61 ± 0.08	1.70 ± 0.15	1.59 ± 0.23	0.03	0.11	0.90	0.87

Note: Results are expressed as mean ± standard error, n = 6. ^a,b^ Peer data with different shoulder labels (lowercase English letters) indicate significant differences (*p* < 0.05).

**Table 2 ijms-26-02045-t002:** Effects of DON and detoxifier on serum biochemical indices of piglets.

Item	Treatment				SEM	*p* Value		
	CON	Det	DON	DON + Det		Det	DON	Det × DON
AST (U/L)	69.98 ± 17.08 ^b^	61.14 ± 16.45 ^c^	88.30 ± 4.86 ^a^	72.31 ± 11.65 ^b^	2.74	0.04	0.01	0.46
ALT (U/L)	59.21 ± 5.96 ^b^	57.19 ± 12.33 ^b^	78.14 ± 16.11 ^a^	54.73 ± 10.30 ^b^	2.40	0.02	0.10	0.04
ALB (g/L)	19.30 ± 2.19	19.25 ± 1.73	17.42 ± 1.67	18.50 ± 2.49	0.42	0.54	0.13	0.51
GLB (g/L)	33.77 ± 4.76	30.97 ± 5.62	33.53 ± 4.20	33.74 ± 5.36	1.02	0.54	0.54	0.47
TP (g/L)	53.06 ± 3.81	54.54 ± 2.02	50.95 ± 5.65	52.24 ± 6.86	1.01	0.50	0.29	0.96

Note: Results are expressed as mean ± standard error, n = 6. ^a–c^ Peer data with different shoulder labels (lowercase English letters) indicate significant differences (*p* < 0.05).

**Table 3 ijms-26-02045-t003:** Effects of DON and detoxifier on serum immunoglobulins of piglets.

Item (g/L)	Treatment				SEM	*p* Value		
	CON	Det	DON	DON + Det		Det	DON	Det × DON
IgA	1.21 ± 0.04	1.17 ± 0.09	1.10 ± 0.29	1.19 ± 0.06	0.03	0.73	0.52	0.33
IgM	0.86 ± 0.03	0.86 ± 0.07	0.81 ± 0.04	0.78 ± 0.07	0.01	0.32	0.62	0.16
IgG	8.73 ± 0.29 ^a^	8.60 ± 0.43 ^a^	7.51 ± 0.64 ^c^	8.10 ± 0.51 ^b^	0.10	0.26	<0.01	0.09

Note: Results are expressed as mean ± standard error, n = 6. ^a–c^ Peer data with different shoulder labels (lowercase English letters) indicate significant differences (*p* < 0.05).

**Table 4 ijms-26-02045-t004:** Effects of DON and detoxifier on serum antioxidant indices of piglets.

Item	Treatment				SEM	*p* Value		
	CON	Det	DON	DON + Det		Det	DON	Det × DON
CAT (U/mL)	42.46 ± 1.57 ^a^	37.36 ± 1.95 ^b^	34.36 ± 1.27 ^c^	39.63 ± 2.97 ^b^	0.42	0.92	<0.01	<0.01
SOD (U/mL)	81.07 ± 3.06 ^a^	77.56 ± 2.15 ^a^	71.52 ± 4.64 ^b^	77.92 ± 2.99 ^a^	0.68	0.30	<0.01	<0.01
GSH-PX (U/mL)	169.97 ± 7.07 ^a^	152.38 ± 8.20 ^b^	143.46 ± 6.04 ^c^	153.38 ± 12.12 ^b^	1.77	0.29	<0.01	<0.01
MDA (nmol/mL)	2.78 ± 0.40 ^b^	2.76 ± 0.46 ^b^	3.99 ± 0.24 ^a^	3.15 ± 0.41 ^b^	0.08	0.01	<0.01	0.02

Note: Results are expressed as mean ± standard error, n = 6. ^a–c^ Peer data with different shoulder labels (lowercase English letters) indicate significant differences (*p* < 0.05).

**Table 5 ijms-26-02045-t005:** Effects of DON and detoxifier on serum inflammatory factors of piglets.

Item (pg/mL)	Treatment				SEM	*p* Value		
	CON	Det	DON	DON + Det		Det	DON	Det × DON
TNF-α	58.49 ± 6.47 ^b^	67.50 ± 2.97 ^a^	70.26 ± 6.18 ^a^	62.95 ± 4.68 ^ab^	1.01	0.70	0.11	<0.01
IL-4	8.43 ± 0.54 ^a^	8.15 ± 0.78 ^a^	7.18 ± 0.20 ^b^	7.96 ± 0.45 ^a^	0.11	0.26	<0.01	0.02
IL-10	19.96 ± 1.06 ^a^	15.65 ± 1.03 ^ab^	13.33 ± 2.28 ^c^	15.13 ± 0.77 ^b^	0.29	0.68	<0.01	0.01
IL-12	42.52 ± 2.27 ^c^	47.66 ± 4.94 ^b^	54.81 ± 3.95 ^a^	47.95 ± 4.47 ^b^	0.82	0.61	<0.01	<0.01
IL-2	283.36 ± 20.27 ^c^	299.23 ± 38.11 ^bc^	358.53 ± 15.44 ^a^	319.71 ± 13.58 ^b^	4.88	0.25	<0.01	0.01
IL-6	133.39 ± 5.54 ^c^	140.24 ± 3.85 ^b^	164.42 ± 5.08 ^a^	148.61 ± 12.88 ^b^	1.57	0.17	<0.01	<0.01

Note: Results are expressed as mean ± standard error, n = 6. ^a–c^ Peer data with different shoulder labels (lowercase English letters) indicate significant differences (*p* < 0.05).

**Table 6 ijms-26-02045-t006:** Effects of DON and detoxifier on serum LPS and DAO of piglets.

Items	Treatments				SEM	*p* Value		
	CON	Det	DON	DON + Det		Det	DON	Det × DON
LPS (EU/mL)	0.18 ± 0.02 ^b^	0.22 ± 0.02 ^b^	0.43 ± 0.16 ^a^	0.21 ± 0.02 ^b^	0.02	0.02	<0.01	<0.01
DAO (U/mL)	1.35 ± 0.13 ^b^	1.31 ± 0.23 ^b^	1.84 ± 0.40 ^a^	1.42 ± 0.16 ^b^	0.05	0.03	<0.01	0.07

Note: Results are expressed as mean ± standard error, n = 6. ^a,b^ Peer data with different shoulder labels (lowercase English letters) indicate significant differences (*p* < 0.05).

**Table 7 ijms-26-02045-t007:** Composition and nutrient levels of experimental diets (%, as-fed basis).

Ingredients	CON	Det	DON	DON + Det
Toxin-free corn	72.96	72.80	0.00	0.00
Soybean meal	18.14	17.05	21.17	21.35
Toxin corn	0.00	0.00	72.65	72.08
Corn gluten meal	2.70	3.55	0.00	0.00
Soybean oil	2.34	2.50	2.35	2.54
Dicalcium phosphate	1.13	1.12	1.12	1.12
Limestone	0.76	0.78	0.75	0.75
Salt	0.35	0.35	0.35	0.35
Vitamin–mineral premix ^a^	1.00	1.00	1.00	1.00
Composite detoxifier	/	0.20	/	0.20
L-Lysine HCl	0.40	0.42	0.35	0.35
DL-Methionine	0.06	0.06	0.09	0.09
L-Threonine	0.13	0.13	0.14	0.14
L-Tryptophan	0.03	0.04	0.03	0.03
Nutrient levels				
Dry matter	88.70	89.10	88.63	89.05
Crude protein	15.67	15.74	15.82	15.62
Gross energy, MJ/kg	16.54	16.43	16.61	16.59
Calcium	0.65	0.68	0.70	0.66
Total phosphorus	0.63	0.63	0.62	0.64
Metabolizable energy, MJ/kg ^b^	13.72	13.72	13.72	13.72

^a^ Vitamin and mineral premix provided the following per kg of diet: vitamin A, 12,000 IU; vitamin D_3_, 2000 IU; vitamin E, 24 IU; vitamin K_3_, 2 mg; vitamin B_12_, 24 µg; vitamin B_2_, 6 mg; vitamin B_1_, 2 mg; vitamin B_5_, 20 mg; vitamin B_6_, 3 mg; niacin acid, 30 mg; choline chloride, 0.4 mg; folic acid, 3.6 mg; biotin, 0.1 mg; Mn, 40 mg (as manganese oxide); Fe, 96 mg (as ferrous sulphate); Zn, 120 mg (as zinc oxide); Cu, 8 mg (as copper sulphate); I, 0.56 mg (as ethylenediamine dihydroiodide); and Se, 0.4 mg (as sodium selenite). ^b^ Metabolizable energy was calculated.

## Data Availability

The datasets supporting this article are available from the corresponding author on reasonable request.

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
