# Peer review of "Effects of Deoxynivalenol Detoxifier on Growth Performance, Blood Biochemical Indices, and Microbiota Composition of Piglets"

_ijms, 2025, doi:10.3390/ijms26052045_

Round 1
Reviewer 1 Report
Comments and Suggestions for Authors
This study highlights the harm of Deoxynivalenol (DON) in pigs and the challenge of its removal from feed. The authors propose a compound detoxification consisting of probiotics, degraders, and adsorbents, effectively reducing DON's adverse effects on piglet growth, serum biochemical induces, and intestinal health. This is a topic of interest to researchers in related fields, but the paper needs some improvement before acceptance for publication. My comments are as follows:
- In lines 122-124, the authors describe the composition of the composite detoxifier. Please clarify how this particular compound was chosen? Are there any references or studies that support the selection of these specific components?
- In Table 1, it is mentioned that the CON and Det groups were fed corn gluten meal, while the DON and DON+Det groups were not. Could you explain why corn gluten meal was included in the diet of the CON and Det groups, but not in the DON and DON+Det groups? Additionally, soybean meal was used at a higher level in the DON and DON+Det groups. Could you clarify the reasoning behind this difference?
- In sections 3.1-3.7 of the results, only two comparisons are presented, namely CON vs. DON and DON vs. DON+Det. Could additional comparisons, such as DON+Det vs. Det, be included to further demonstrate the effect of the detoxifier?
- In sections 3.1-3.7 of the results, in describing growth and serum biochemical indicators, the author notes significate differences but lacks specific numerical values (e.g., means or ranges) and discussion of biological significance. Including these details would enhance clarity and depth.
- In sections 3.2-3.5 of the results, the text describes the figures without properly referencing their corresponding numbers or providing detailed image information. For example, the authors only mention 'Figure' in the first sentence; it would be clearer to specify the figure and part, such as 'Figure 1A,' at the appropriate point in the text. Please check all.
- In section 3.2 of the results, the text compares the DON+Det and DON groups, but Figure 1B shows significances for Det vs. DON and Det vs. DON+Det, creating inconsistency between the text and figure. Similar issues occur in Figures 3 and 4 and should be corrected.
- You need to follow a consistent format for the figure legends. For example, Figure 1 first describes the details of the image, followed by an overview, while Figure 6 starts with an overview and then describes the image details. The format used in Figure 6 is typically preferred.
8. In the figure descriptions, it is necessary to include figure legends. For example, in Figure 8, the legend should specify that orange represents CON, blue represents Det, etc. Please check the entire manuscript to ensure that all figures include appropriate legends.
Comments on the Quality of English LanguageThe English could be improved to more clearly express the research.
Author Response
Dear reviewer,
Thank you very much for your comments and professional advice. These opinions help to improve academic rigor of our article. Based on your suggestion and request, we have made corrected modifications on the revised manuscript. Meanwhile, the manuscript had be reviewed and edited by language services. We hope our work can be improved again. Furthermore, we would like to show the details as follow:
Comments 1: In lines 122-124, the authors describe the composition of the composite detoxifier. Please clarify how this particular compound was chosen? Are there any references or studies that support the selection of these specific components?
Response 1: Thank you for your valuable comments. In selecting the composition of the deoxynivalenol detoxifier, we relied on the outcomes of previous in vitro studies that have demonstrated its efficacy in DON detoxification. Furthermore, we consulted relevant literature and research to inform our ingredient selection. For instance, ingredients such as Pediococcus pentose , Saccharomyces cerevisiae , Lactobacillus plantarum , glucose oxidase, Eucommia ulmoides leaf and adsorbent (modified sodium humate) have been shown to effectively adsorb and detoxify DON. The combination of these ingredients exhibited a synergistic effect in our in vitro experiments, thereby enhancing the overall detoxification efficacy (The detailed results are presented as depicted in the following figure ). Additionally, we drew upon other research focused on optimizing detoxification through the adsorption properties and chemical stability of various materials, which provided theoretical support for our composition choices.The relevant references have been cited in the introduction, such as references 8, 9, 10, and 16, etc.
Note: Treat1: Pediococcus pentose+glucose oxidase+adsorbent+Eucommia ulmoides leaf
Treat2: Saccharomyces cerevisiae+glucose oxidase+adsorbent+Eucommia ulmoides leaf
Treat3: Lactobacillus plantarum+glucose oxidase+adsorbent+Eucommia ulmoides leaf
Treat4: Pediococcus pentose+Saccharomyces cerevisiae+glucose oxidase+adsorbent+Eucommia ulmoides leaf
Treat5: Pediococcus pentose+Lactobacillus plantarum+glucose oxidase+adsorbent+Eucommia ulmoides leaf
Treat6: Saccharomyces cerevisiae+Lactobacillus plantarum+glucose oxidase+adsorbent+Eucommia ulmoides leaf
Treat7: Pediococcus pentose+Saccharomyces cerevisiae+Lactobacillus plantarum+glucose oxidase+adsorbent+Eucommia ulmoides leaf
Comments 2: In Table 1. it is mentioned that the CON and Det groups were fed corn gluten meal. while the DON and DON+Det groups were not. Could you explain why corn gluten meal was included in the diet of the CON and Det groups, but not in the DON and DON+Det groups? Additionally, soybean meal was used at a higher level in the DON and DON+Det groups. Could you clarify the reasoning behind this difference?
Response 2: Thank you for your valuable comments. Composition and nutrient levels of experimental diets were shown in Table 3. Regarding the differences in the use of corn gluten meal and soybean meal in feed formulations, we explain as following: In this study, CON and Det groups used Toxin-free corn, which has a low crude protein content (CP 7.6%). DON and DON+Det groups used Toxin has a higher crude protein content (CP 7.97%) . In order to maintain a similar proportion of corn in each group, while ensuring consistent crude protein and metabolizable energy levels, which could not be achieved by adjusting the ratio of soybean meal alone, so we added corn gluten meal which has a higher protein content (CP 59.2%) to the CON and Det groups. In the DON and DON+Det groups, due to the high crude protein content of the corn used itself, we adjusted the metabolizable energy and other nutrients by increasing the amount of soybean meal to bring it in line with the other two groups. This adjustment is based on the principles of protein and energy balance commonly found in feed formulation design. Through the above adjustment, we ensured the nutritional balance of each group's feed and met the requirements of the experimental design. If further information or data support is required, we will be happy to provide it.
Comments 3: In sections 3.1-3.7 of the results, only two comparisons are presented, namely CON vs. DON and DON vs. DON+Det, Could additional comparisons, such as DON+Det vs. Det. beincluded to further demonstrate the effect of the detoxifier?
Response 3: Thank you for your suggestion. We have revised the result presentation section of the manuscript, and added comparisons between the Det group and the DON+Det group, which have been highlighted in red.
Comments 4:In sections 3.1-3.7 of the results, in describing growth and serum biochemical indicators, the author notes significate differences but lacks specific numerical values (e.g., means orranges) and discussion of biological significance. Including these details would enhance clarity and depth.
Response 4: Thank you for your suggestion.We have changed the presentation of the results regarding the biochemical indices of piglet serum, antioxidant indices, serum immunoglobulins, and inflammatory factors in the original manuscript from figures to tables. This is to enable you and the editor to more clearly and intuitively observe the comparisons and differences among the data. The biological mechanisms corresponding to different indices have been described in greater detail in the discussion section.
Comments 5:In sections 3.2-3.5 of the results, the text describes the fiaures without properly referencing their correspondina numbers or providing detailed image information, For example. The authors only mention 'Figure' in the first sentence: it would be clearer to specify the figure and part, such as 'Figure 1A,' at the appropriate point in the text. Please check all.9
Response 5: Thank you for your suggestion. We have revised the result description section of the manuscript, and have referenced the corresponding numbers of the cited documents in detail, which have been highlighted in red.
Comments 6: In section 3.2 of the results, the text compares the DON+Det and DON groups, but Figure 1B shows significances for Det vs. DON and Det vs. DON+Det, creating inconsistency between the text and figure. Similar issues occur in Fiqures 3 and 4 and should be corrected.
Response 6: Thank you for your suggestion. We have revised the result presentation section of the manuscript to ensure consistency between the result descriptions and the information presented in the tables.
Comments 7: You need to follow a consistent format for the figure legends. For example, Figure 1 first describes the details of the image, followed by an overview, while Figure 6 starts with an overyiew and then describes the image details, The format used in Figure 6 is typically preferred.
Response 7: Thank you for your suggestion. We have modified the format of all the figures to be consistent, that is, to provide an overview first and then describe the details of the tables , which have been highlighted in red.
Comments 8: In the figure descriptions, it is necessary to include figure legends. For example, in Figure 8, the legend should specify that orange represents CON, blue represents Det, etc. Please check the entire manuscript to ensure that all figures include appropriate legends.
Response 8: Thank you for your valuable comments. We have already annotated the colors in Figure 4 (The original Figure 8) and the groups they represent, and marked them in red.
Comments: The English could be improved to more clearly express the research.
Response: Thank you for your valuable comments.We have checked and corrected the English expressions throughout the manuscript, and the corrections are marked in red in the revised version.
Thank you very much for your attention and time. Look forward to hearing from you.
Yours sincerely,
Lu-yao Zhang
23 Feb. 2025

Reviewer 2 Report
Comments and Suggestions for Authors
Introduction: The introduction effectively outlines the issue of DON contamination in animal feed.
Methodology: The methodology section requires clarification regarding the timing of animal weighing in relation to feeding.
Discussion: The discussion adeptly integrates the findings of this study with existing literature, emphasizing the implications of the results for pig health and performance..
Conclusion: The conclusion could be enhanced by suggesting potential avenues for future research.
Author Response
Dear reviewer,
Thank you very much for your comments and professional advice. These opinions help to improve academic rigor of our article. Based on your suggestion and request, we have made corrected modifications on the revised manuscript. Meanwhile, the manuscript had be reviewed and edited by language services. We hope our work can be improved again. Furthermore, we would like to show the details as follow:
Comments 1: The methodology section requires clarification regarding the timing of animal weighing in relation to feeding.
Response 1: Thank you for your suggestion. We have provided a detailed description of the timing arrangement between piglet weighing and feeding times, and the revised content has been highlighted in red in Section 4.5. The details as follows:
Before:
The piglets from each replicate were weighed at the beginning and end of the experimental period, and the feed consumption of the piglets was recorded as well to determine average daily gain (ADG), average daily feed intake (ADFI), and feed-to-gain ratio (F/G).
Now:
At days 1, 14, and 28 of piglet feeding, the body weight (BW) of the piglets was measured after a 12-hour fasting period with continuous access to water. The feed intake of the piglets was recorded on a per cage basis, and the average daily gain (ADG), average daily feed intake (ADFI), and feed-to-gain ratio (F/G) were calculated for the periods of 1-14 days, 15-28 days, and the entire experimental period.
Comments 2:The conclusion could be enhanced by suggesting potential avenues for future research.
Response 2: Thank you for your suggestion. We have already added potential directions for future research in the original conclusion and highlighted them in red in the text.The details as follows:
Before:
The compound detoxifier used in this study can regulate flora homeostasis by increasing the abundance of beneficial bacteria and decreasing the abundance of harmful bacteria, so as to alleviate the adverse effects of DON contaminated diet on growth performance, antioxidant function and inflammatory response of piglets.
Now:
The compound detoxifier used in this study can regulate flora homeostasis by increasing the abundance of beneficial bacteria and decreasing the abundance of harmful bacteria, so as to alleviate the adverse effects of DON contaminated diet on growth performance, antioxidant function and inflammatory response of piglets. Looking ahead, further research could explore the optimal formulation of this composite detoxifier and its application effects across different growth stages and feeding environments, aiming to maximize its detoxifying efficacy while reducing production costs. Additionally, conducting in-depth studies on the mechanism of action of this detoxifier, particularly its regulatory pathways at the molecular level, will provide theoretical foundations and technical support for the development of novel and highly efficient feed detoxifiers.
Thank you very much for your attention and time. Look forward to hearing from you.
Yours sincerely,
Lu-yao Zhang
23 Feb. 2025
